# Assessing Farmers' Typologies of Perception for Adopting Sustainable Adaptation Strategies in Bangladesh

**Abu Reza Md. Towfiqul Islam** [1,2,3], **Md. Hasanuzzaman** [1], **Mahmud Jaman** [1], **Edris Alam** [4,5], **Javed Mallick** [6], **G. M. Monirul Alam** [7], **Md. Abdus Sattar** [8] **and Kuaanan Techato** [2,3,*]

1   Department of Disaster Management, Begum Rokeya University, Rangpur 5400, Bangladesh; towfiq_dm@brur.ac.bd (A.R.M.T.I.); md.hasanuzaaman.bp@gmail.com (M.H.); mahmudjaman11@gmail.com (M.J.)
2   Faculty of Environmental Management, Prince of Songkla University, Songkhla 90112, Thailand
3   Environmental Assessment and Technology for Hazardous Waste Management Research Center, Faculty of Environmental Management, Prince of Songkla University, Songkhla 90110, Thailand
4   Faculty of Resilience, Rabdan Academy, Abu Dhabi 22401, United Arab Emirates; ealam@ra.ac.ae
5   Department of Geography and Environmental Studies, University of Chittagong, Chittagong 4331, Bangladesh
6   Department of Civil Engineering, King Khalid University, Abha 61411, Saudi Arabia; jmallick@kku.edu.sa
7   Department of Agribusiness, Bangabandhu Sheikh Mujibur Rahman Agricultural University, Gazipur 1706, Bangladesh; gmmonirul79@gmail.com
8   Department of Disaster Risk Management, Patuakhali Science and Technology University, Patuakhali 8602, Bangladesh; abdus.sattar@pstu.ac.bd
*   Correspondence: kuaanan.t@psu.ac.th

**Abstract:** The implementation of sustainable adaptation strategies (SASs) is crucial to mitigate climate change impact as well as reduce the loss of natural disasters and increase agricultural crop production. However, current policies and programs based on agricultural incentives are mostly inadequate to increase SASs practices at the farm level. Hence, a deeper understanding of farmers' 'perceived typologies to the environmental issue and climate change' is necessary for implementing SASs to enhance farmers' ability to adapt at the farm level. This research intends to demarcate farmers in various categories, according to their perceptions on environmental and climate change issues in the northern part of Bangladesh. Principal component analysis (PCA) and cluster analysis (CA) were employed to analyze the survey data collected from 501 households in the study area. Farmers were clustered into three types, 'Ecocentric', 'Worried', and 'Anthropocentric', based on their perceived knowledge regarding environmental issues and climate change, which guides the adoption of SASs. The 'Worried' cluster showed a high sense of perceived risk of climate change and a significant positive association with the adoption of SASs. By contrast, 'Ecocentric' and 'Anthropocentric' groups showed a low sense of awareness of climate change and a significant negative association with the adoption of SASs. The findings can assist policymakers in promoting the adoption of SASs based on the farmers' cluster and thus enhance their resilience.

**Keywords:** agricultural crop production; Bangladesh; pro-environmental behaviors; ecocentric; sustainable adaptive measures

## 1. Introduction

The agricultural sector is tackling enormous challenges, such as environmental degradation and climate change impact, which are expected to decrease agricultural crop production. Despite this reduction, the agricultural sector will need to produce more than 50 percent more crops in the mid-century era than it did in 2012 [1]. Over the years, climate change impact has been broadly documented by scientific societies and decision-makers as one of the most pressing environmental challenges [2,3]. Agriculture is affected by climate change and is closely associated with climate variables [4]. However, climate-induced

disasters in the agricultural sector are the major limiting factors for ensuring food security in many developing countries, including Bangladesh. There is a growing pressure to meet the food demand of the escalating population [5], and thus improvement in the agricultural sectors in Bangladesh is urgent [6–10].

Climate change challenges in agricultural sectors necessitate the use of innovative sustainable adaptation strategy (SASs) methods to avoid climate change impacts while simultaneously improving crop yield. The FAO has widely recognized the use of SASs against climate change [11]. A myriad number of recent works have reported that the adoption of SASs at the farm level is an appropriate option in meeting these challenges [12–17]. Some of the SASs is available to farmers who can save water by using water-saving irrigation techniques, increasing water storage, and using water efficiently [18,19]. Using organic fertilizer, replacing crop varieties, and changing cropping patterns are examples of SASs that are likely to concurrently increase crop production. Farmers' economic gains will improve as a result of implementing appropriate SASs to mitigate climate change effects and conserve soil and water [20]. Despite these advantages, the implementation of SASs still remains low in many developing countries, including Bangladesh [21]. For example, recent agricultural census data revealed that less than 10% of farmers have adopted changing cropping patterns in Bangladesh [22]. The reason that elucidates the low adoption rate of SASs at the farm level is evident: farmers are driven by economic motivations. Farmers have been assisted by the government and non-governmental organizations, particularly in Europe, through financial incentives [23]. However, there is limited evidence in Bangladesh, where policy based on incentives has mostly been inadequate to motivate farmers to adopt the SASs [24,25]. Hence, policy and programs that focus on economic incentives should be revised in Bangladesh to enhance the implementation of SASs by coupling them with non-economic incentives.

Farmers' perception can be classified based on typology to have a better understanding of the factors that affect the implementation of adaptation strategies [26]. Each category comprises a specific group of farmers that hold the same perceptions. This categorization can efficiently help to provide the heterogeneity motivations of the farmers associated with a specific behavior [27]. Several studies have revealed the typology method to investigate farmers' adaptive efficacy to climate change [28–30] and on farmers' perceived environmental values [31]. Nevertheless, the role of the environmental issue on a farmer's perceived typologies is still less investigated, possibly due to their new addition in the existing literature. There is an urgent need to explore not only how farmers perceive environmental issues and climate change, but also how they appraise their sustainable adaptive strategies (SASs). Novel SASs would efficiently adapt to environmental value and climate change if the farmer is willing to plan to take precautionary actions and lessen the existence of mal-adaptation practices [32]. For instance, a study was conducted on farmers' perception with a focus on non-economic factors such as land ownership, the change creation, and recognition, all of which affect farmers' behavior and motivates the categorization of typologies [33]. It has been reported that understanding farmers' socioeconomic characteristics and perceptions is essential to enhance the adoption of SASs [34–37]. Despite the importance of using the typology technique to improve farmers' adaptive efficacy to climate change, there are few studies that look at farmers' behavior based on typology [38–41].

Understanding farmer perceptions of environmental problems and climate change is essential for implementing long-term adaptive measures to improve farmers' ability to adjust at the farm level. To the authors' knowledge, no study has looked at farmers' perceived typologies in the context of an environmental problem and climate change in Bangladesh, particularly in the northern region. We hypothesized that economic factors are the key drivers of farmer behavioral change. This research aims to close this gap in the literature by classifying farmers in northern Bangladesh into different typologies based on their perception of environmental problems and climate change issues which can help farmers in developing sustainable adaptation practices. This paper may be helpful in

determining which interferences are sufficient to enhance adaptive efficacy and also cope with climate change for each cluster of farmers.

## 2. Materials and Methods

### 2.1. Study Area Description

The present study was carried out in the northern region of Bangladesh, which encompasses an area of about 20,000 km² [22]. The northern region is characterized by the sub-humid zone and the imbalance of soil moisture conditions [10]. This area is one of the agricultural crop production hubs in Bangladesh. This region is characterized by a sub-tropical monsoonal climate with a mean temperature range from 17–20 °C in the winter and 26.9 to 31.1 °C in the summer season [42], 1100–1329 mm of annual rainfall, 600–800 mm of evapotranspiration, and 60–80% of relative humidity [43]. With irregular patterns of rainfall distribution, rainfall occurs mainly in the monsoon summer season (July to October), which enhances the intensity and frequency of rainstorms and the probability of soil erosion [42]. Furthermore, the amount of rainfall is comparatively lower in this region than in other parts of the country, leading to a meteorological drought risk, which is unfavorable for agricultural production. Thus, it is the most fragile area in the aspect of the agro-ecological point of view.

Based on the consultation with the expert seven districts of northern Bangladesh and then one upazila from each district, one village from each upazila was selected for this study. The selected districts were *Panchagarh, Thakurgaon, Dinajpur, Nilphamari, Rangpur, Lalmonirhat and Kurigram with each villags of selected upazilas* of Rangpur division under northern Bangladesh (Figure 1).

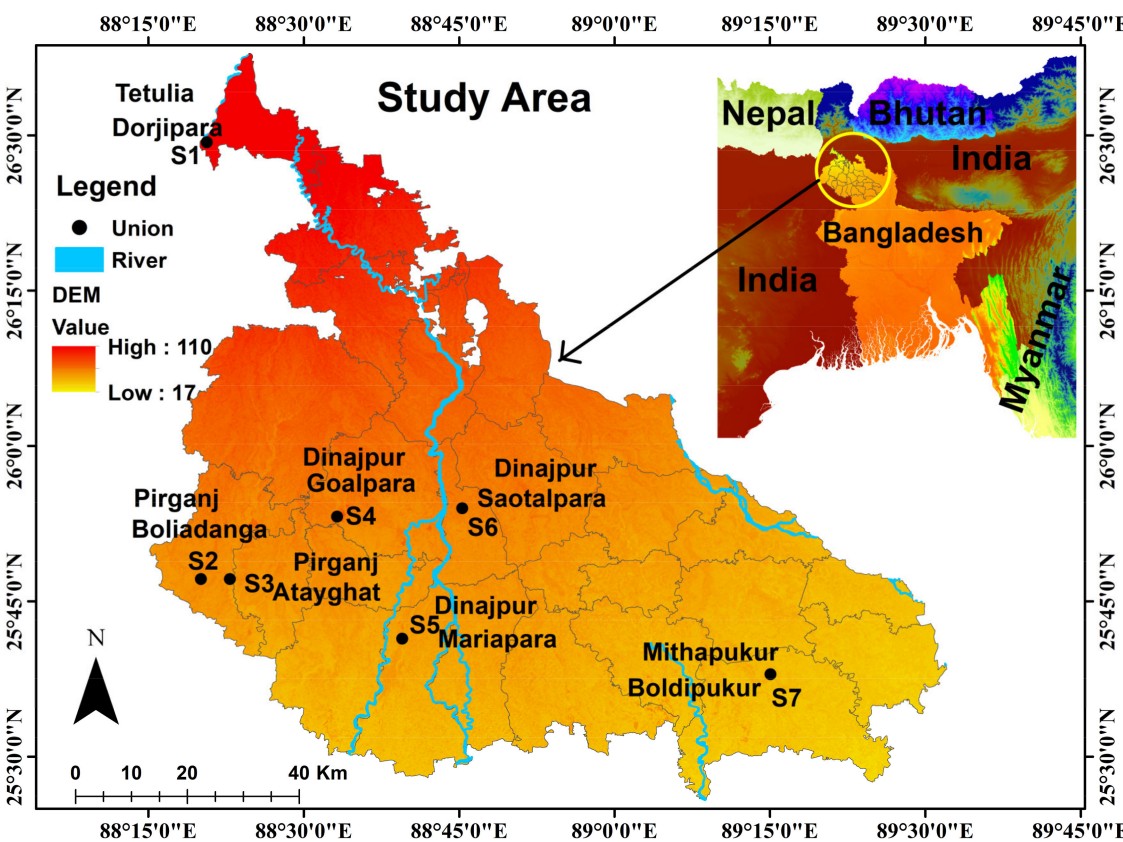

**Figure 1.** Location map showing the study area, where black solid circles represent the study sites from where survey data were taken from northern part of Bangladesh.

### 2.2. Sampling and Design of the Questionnaire

A stratified random sampling technique was employed to collect the data. First, 7 districts were chosen that had the typical agricultural practices. Second, 7 upazilas from these districts were chosen. Third, one village from each upazilla was randomly selected for data collection.

The statistical technique developed by [44,45] was used to determine the sample size. This technique was applied by several research scholars [46–48] for estimating household sample size. Thus, data were collected from 501 farmers for this study.

The household head (either male or female) was the respondent for the survey. Information was obtained from each household head, who is involved in farming activities, through a structured questionnaire survey. Before the screening questions were asked, a short description of this survey was presented to farmers. Primary data were collected from the household's head by employing face-to-face interviews.

It is worth mentioning that this questionnaire has two parts: the first part was focused on collected information on the farmer's profile from the agricultural extension office. Additionally, farmers revealed which SASs they had applied on their farms. The second part contained 20 statements determined in a five-point Likert scale. After an extensive review of the literature, 20 adoption strategies/options were selected (Supplementary Table S1). The farmers were asked to rate the strategy for how effective they believed the strategy would be as a sustainable adoption. On the other hand, we selected 73 statements according to causes, impact, and prevention of climate change. Then, we obtained 18 attitudinal statements after performing principal component analysis (PCA). Both attitudinal statements are related to climate change. So, both of them give a proper perception about how climate change affects agriculture and what strategies they prefer. It is important to note that 18 statements made the questionnaire based on a five-point Likert scale based on environmental issues, causes, and climate change impacts. These statements were amended from the existing literature, e.g., [30,49,50], and the feedback of ten Bangladeshi experts in the field of agro-adaptation practices and farm-level management. The 18 statements are outlined in Table 1.

**Table 1.** Factor loading values of attitudinal statements come from principal component analysis (PCA).

| Serial No. | Attitudinal Statements | Components [a] | | | Communalities |
|:---:|:---:|:---:|:---:|:---:|:---:|
| | | 1 | 2 | 3 | |
| 1 | Does air pollution cause climate change? | 0.67 | | | 0.49 |
| 2 | Does soil pollution cause climate change? | 0.91 | | | 0.83 |
| 3 | Does water pollution cause climate change? | 0.94 | | | 0.89 |
| 4 | Does plastic pollution cause climate change? | 0.98 | | | 0.95 |
| 5 | Does using pesticides cause climate change? | 0.93 | | | 0.89 |
| 6 | Do mills/industry/car smoke/$CO_2$ cause climate change? | 0.89 | | | 0.88 |
| 7 | Does unawareness of the negative consequences of climate change due to manmade activities cause climate change? | 0.96 | | | 0.94 |
| 8 | Does refrigeration cause climate change? | 0.80 | 0.50 | | 0.89 |
| 9 | Do other human influenced factors cause climate change? | | 0.66 | | 0.65 |
| 10 | Is the annual mean temperature changing due to climate change? | | 0.84 | | 0.73 |
| 11 | Has the annual mean precipitation been changed due to climate change? | | 0.76 | | 0.61 |
| 12 | Has the seasonal variation in terms of duration and starting time been changed due to climate change? | | 0.61 | | 0.37 |
| 13 | Do drought and flood occur frequently due to climate change? | | 0.57 | | 0.36 |
| 14 | Is temperature rising due to climate change? | | 0.58 | | 0.37 |
| 15 | Is solar radiation increasing due to climate change? | | 0.62 | | 0.40 |
| 16 | Is the humidity level high due to climate change? | | 0.69 | | 0.52 |
| 17 | Does deforestation influence the climate change? | | | 0.72 | 0.53 |
| 18 | Do natural factors (Sun's heating imbalance, etc.) influence tclimate change? | | | 0.79 | 0.67 |
| | Cronbach's Alpha | **0.96** | **0.85** | **0.63** | |
| | Eigen values | **7.35** | **3.68** | **1.36** | |
| | % Explained variance | **35.02** | **17.49** | **6.47** | |
| | % Cumulative variance | **35.02** | **52.52** | **58.99** | |

[a] **Factor:** (**1**) Awareness of climate change (**2**) Perceived risk (**3**) Environmental behavior.

The questionnaire datasets were based on a number of focus group discussions (FGDs) amongst field experts on climate and social science background, local agricultural officers, and farmers. The FGDs aimed to acquire more detailed data concerning environmental issues and climate change conditions and adaptation measures and obtain views about the questionnaire design. A pre-test of farmers in the present study region was done to

avoid any uncertainty in these questions and confirm they were reasonable. Based on the feedback from the pre-test, some explanations and amendments were made. Moreover, the interviewers filled in the farmer's name and detailed information, but this personal information was not disclosed due to personal privacy.

Data were collected from November 2018 to January 2019. Among the 501 farmers interviewed, 59% of household heads were male and 41% were female. The highest proportion of respondents (37.2%) was under the age group of 41–50, followed by 51–60 years old (32.2%). About 81.4% of the respondent had achieved primary level education, and the rest of them were beyond that level. The illiteracy rate was 30%. Farmers whose annual agricultural income was USD 701–1050 accounted for 62.8%, while non-agricultural income was USD 351–700 accounted for 8.8%.

Furthermore, this study developed a modified typology-based K-means clustering tool to divide farmers into typology based on similar perception. This analysis required information on the farm household characteristics of each cluster formed and next, the cluster was demarcated into farmers' perceived environmental values and current climate change situations using the concepts provided by [49].

### 2.3. Statistical Analyses

Principal component analysis (PCA) and clusters analysis (CA) were performed by using the Statistical Package for the Social Sciences (SPSS) version 25 and software R version 3.6.3. At first, to reduce the indicators, we performed PCA on causes, consequences, and effect variables that are positively associated with climate change. We excluded two SASs variables because these variables were negatively associated with climate change. For indicator reduction, we used PCA where the Kaiser–Maier–Olkin (KMO) test, as well as Bartlett's sphericity test, were performed for verifying the fittingness of the dataset. The results of KMO > 0.5 (Original value was 0.897) at $p < 0.01$ significance level verify that the data were suitable for PCA analysis in this research [51]. Each factor had each statement belonging to factor loading >0.5, which is similar to [25]. Communalities' values of each statement referred to >0.40 being acceptable for PCA [51]. The main advantage of the PCA is that this technique can help reduce the dataset for keeping inherent data characteristics. The Cronbach's alpha (CRA) was employed to examine the internal variability and consistency of these factor loadings. The CRA values > 0.6 are thought to be suitable in social science studies [18,52]. In addition to this, based on factor scores from PCA, cluster analysis was applied. Ward's hierarchical method was used to identify the possible number of clusters (Figure 2). Since this study is subjective, it needed a research scholar's judgment for elucidating the findings [34]. From the hierarchal clustering analysis (HCA), we got 3 clusters. K-means HCA technique was added in this analysis. The K-mean HCA technique reduces the distances in each cluster center within all the clusters [30]. The benefit of the HCA is that it can help to categorize farmers' perceptions concerning environmental values and climate change. A Kolmogorov–Smirnov test was performed to validate the significant difference among all clusters ($p < 0.05$). Overall, the methodological steps of this study are shown in the flowchart (Figure 3).

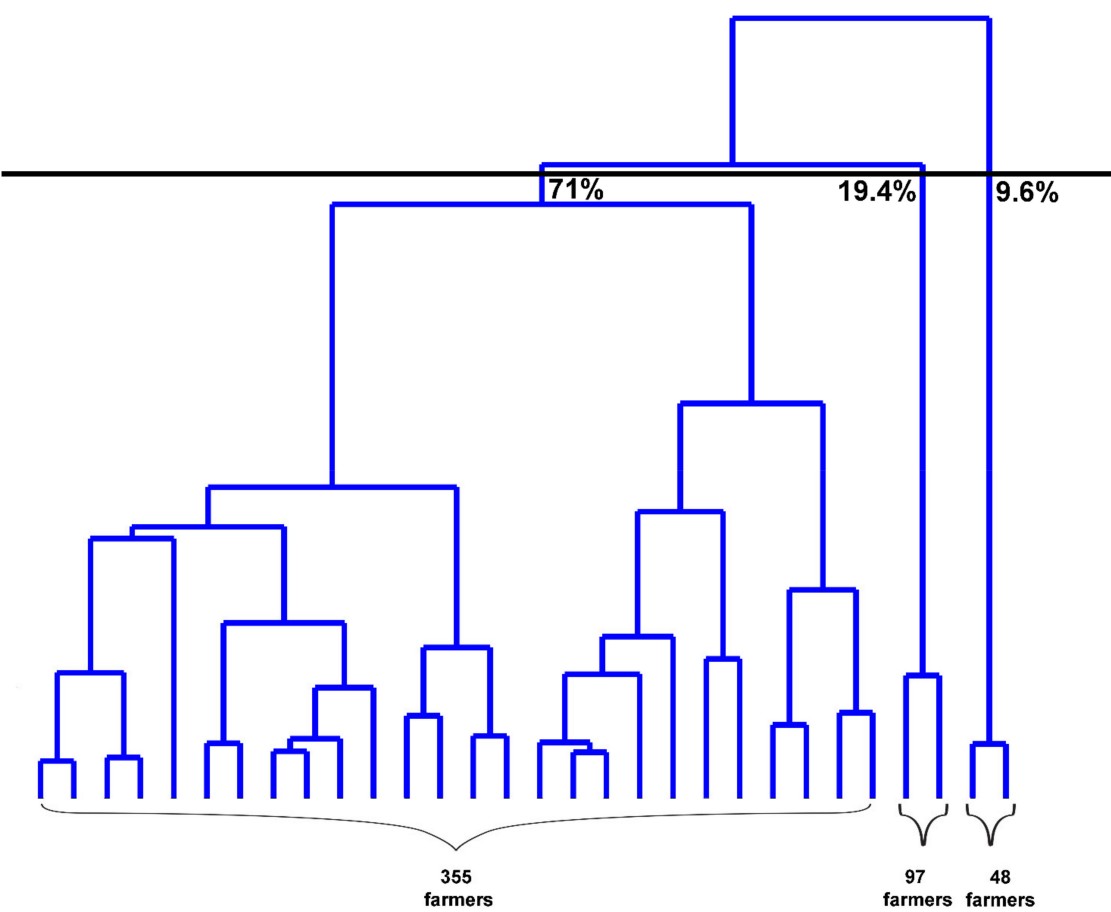

**Figure 2.** Dendrogram cluster analysis showing farmers' typologies in the northern part of Bangladesh.

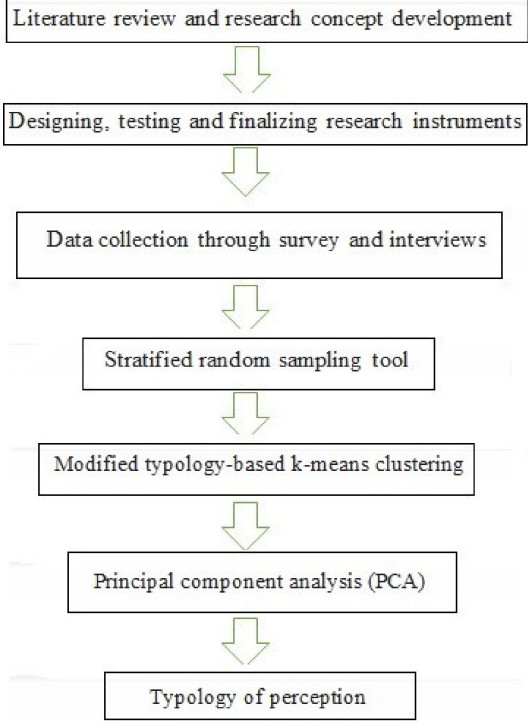

**Figure 3.** Flowchart showing the methodological steps of this study.

## 3. Results

### 3.1. General Characteristics of Each Cluster Farmers'

Table 1 shows the outcomes of PCA. Four factors produced an eigenvalue greater than 1. These factors elucidated 58.99% of the overall variance. According to the attitudinal statements, they were categorized based on each factor as: (1) Awareness of climate change, (2) Perceived risk, and (3) Environmental value. Factor 1 explained 35.021% of the variance (awareness of climate change), factor 2 elucidated 17.498% of the variance (perceived risk), and factor 3 explained 6.472% of the variance (environmental value). Statements 12, 13, and 14 outlined low communalities values. We decided to include them in our analyses because the commonalities value of these statements was more than 0.35. The Cronbach's alpha factor values varied from 0.96 (Awareness of climate change) to 0.63 (Environmental value), and thus, they were fit for further analysis. Different typologies are demonstrated by using the score of the factor loadings from PCA. Using cluster analysis, three typologies were detected in this study (Table 2). Factors scores that were found by performing PCA were used to classify the farmers into various typologies (groups). A radar diagram was prepared to elucidate a graphical illustration of the disparities of each cluster in this study (Figure 3). According to 'typology', the study farmers were categorized into three clusters considering different farmer types as 'Ecocentric', 'Worried', and 'Anthropocentric', respectively.

**Table 2.** Total scores of the final centers of farmer's clusters using K-means clustering technique.

| Factors | Clusters | | |
|---|---|---|---|
| | Ecocentric (*n* = 355) | Worried (*n* = 97) | Anthropocentric (*n* = 48) |
| Awareness of climate change | 0.29 | −0.62 | −0.85 |
| Perceived risk | 0.30 | 0.29 | −2.82 |
| Environmental value | 0.44 | −1.23 | −0.76 |

'Ecocentric' indicates that this class of people had a high perceived high environmental value with a low sense of awareness of climatic knowledge and perceived risk [49] (Cluster 1) (Figure 4a). Though they care about climate change, their main focus is on how the environment is affected. Cluster 1 suggests pro-environmental value, which and represents those more likely to implement appropriate strategies. The farmers in the 'Ecocentric' category have much more concern about environmental values. So, they are likely to take measures to prevent the causes of climate change. On the other hand, the 'Worried' category exhibited the farmers with contrary perspectives in the other clusters, representing the highest sense of perceived risk that can negatively affect many regions of the world (Figure 4b). This was the reason for selecting this name in cluster 2. The 'Worried' (Cluster 2) farmers have less interest in environmental values, as they have the most interest in perceived risk, which suggests a lower likelihood to adopt sustainable practices. They focus on which negative consequences occur due to climate change. Thus, farmers belonging to cluster 2 revealed that environmental issues are exaggeratedly presented. Farmers in the "Anthropocentric" category are driven by the coupling of awareness of climate change and environmental value, which belong to cluster 3 (Figure 4c). They focus on which natural and manmade issues influence climate change. 'Ecocentric' and 'Anthropocentric' both show pro-environmental behaviors, in which features are presented in [49]. It suggests that they are likely to adopt SASs, but several reasons motivate them. 'Ecocentric' farmers value the environment, as they think it needs to be protected because of its essential value (ecological settings, natural resources, etc.). However, 'Anthropogenic' farmers think that the environment should be preserved because it enhances the quality of human life (health facility, human comfort-ability, quality of life, etc.) [30,49].

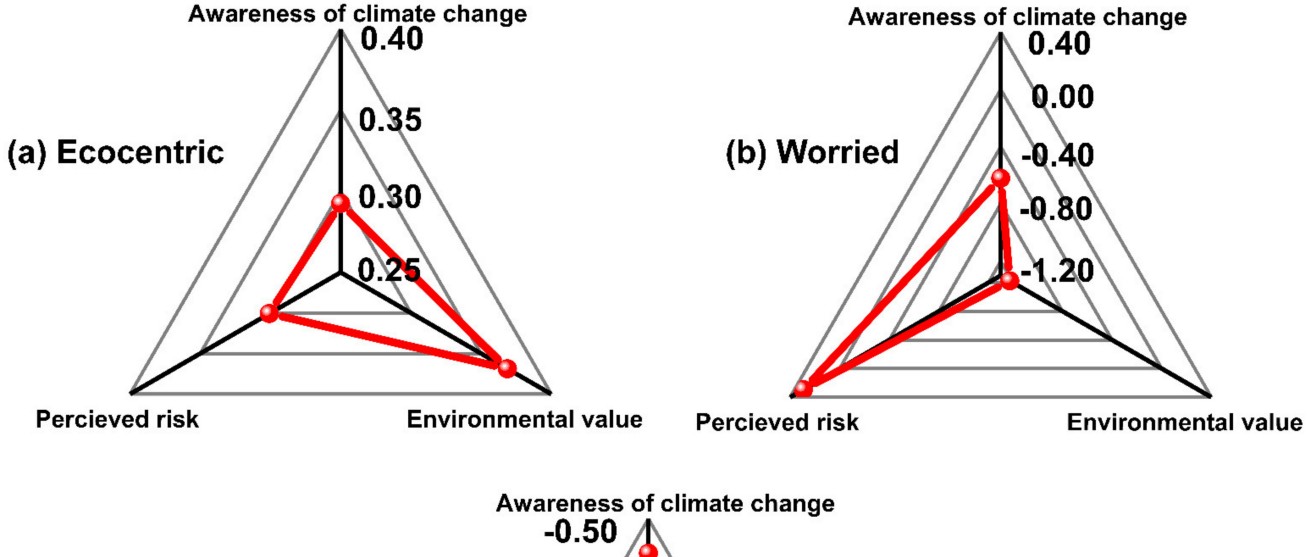

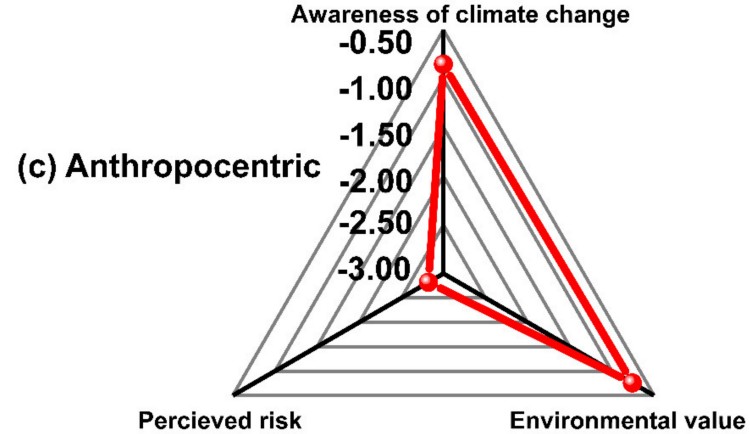

**Figure 4.** Radar diagrams exhibiting the scores of three clusters identified in the study area.

*3.2. Characteristics Features of Farmers' According to Three Clusters*

The general features of the participants are shown in Table 3 and categorized according to each cluster. The K–S test exhibited a difference among the three clusters at a significance level of $p < 0.05$. The 'Ecocentric', 'Worried', and 'Anthropocentric' are the clusters that represent 71%, 19.4%, and 9.6%, respectively, in the analysis. In the 'Ecocentric' category, 355 farmers were included; most of them had lived in that area for more than 30 years, and the farmers were relatively older than those in other clusters. They have more experience than the farmers of other clusters with the highest agricultural earning (USD 1028.44) (Table 3). Farmers in the 'Ecocentric' group are highly educated and earn the lowest non-agricultural income comparted to the remaining two clusters. On the other hand, the highest non-farm (non-agricultural) activity was found in 'Worried' farmers (25.77%) and the second highest was found in 'Ecocentric' farmers (23.66%). By contrast, farmers in the 'Anthropocentric' group took loans that were two times higher than farmers in the 'Ecocentric' group. in the 'Worried' and 'Anthropocentric' categories, 97 and 48 farmers, respectively are included who live in the area same as 'Ecocentric' farmers. Due to the highest number of family members in the 'Anthropocentric' category, the total earnings are the highest out of the clusters (at an average of USD 1015.81 from agricultural and USD 49.78 non-agricultural activity). The average farm size is almost the same in all typologies.

**Table 3.** The mean values of farmers' feature variables of 3 designated typologies [a].

| SL | Farmers' Feature Variables | Ecocentric (n = 355) | Worried (n = 97) | Anthropocentric (n = 48) |
|---|---|---|---|---|
| 2 | Age | 51.49 | 50.81 | 48.92 |
| 3 | Dependency Ratio | 0.85 | 0.81 | 0.79 |
| 4 | Earning Member | 1.21 | 1.16 | 1.19 |
| 5 | Educational status [b] | 1.21 | 1.19 | 1.15 |
| 6 | Total household members | 5.18 | 5.15 | 5.29 |
| 7 | Total annual agricultural income (US Dollar) | 1028.4 | 1002.3 | 1015.8 |
| 8 | Total annual non-agricultural income (US Dollar) | 12.72 | 39.21 | 49.78 |
| 9 | Farm size (ha) | 0.25 | 0.25 | 0.25 |
| 10 | Number of sources for taking loan | 0.34 | 0.26 | 0.19 |

[a] Kolmogorov–Smirnov test was performed demonstrating significant difference among all clusters ($p < 0.05$). [b] 1: Primary education, 2: Secondary education 3: Higher Education.

### 3.3. Perception on SASs According to Farmers' Typologies

All three types of farmers' perceptions about SASs have been demonstrated in Table 4. 'Anthropocentric' farmers' have comparatively less interest in the adoption of SASs than 'ecocentric' farmers, where the highest interest is found in using organic fertilizers (average value of 3.7). Surprisingly, it was found that both 'Ecocentric' and 'Worried' farmers have the most interest (mean value is more than 4 out of 5) in adopting the same eight SASs, which include afforestation, conserving soil, using organic fertilizers, replacing the crop varieties, increasing irrigation, mango farming, livestock farming, and public awareness (at an average of 4.3, 4.2, 4.1, 4.2, 4.1, 4.1, 4.2, and 4.3 for 'Ecocentric' and 4.2, 4.2, 4.1, 4.1, 4.1, 4.2, 4.2, and 4.3 for 'Worried', respectively). The 'Anthropocentric' farmers had the highest interest in SASs of organic fertilizers and preparing for extreme weather. The highest interest in SASs based on afforestation and public awareness was found in the 'Ecocentric' group. Some SASs have a lower interest due to the features of the farmers. For example, they are often distributed in the floodplain region and do not adopt the change of disease and insect pests. Although some farmers have expertise in cultivating rice crops, they then prefer to adopt livestock farming.

**Table 4.** Descriptive analysis among farmers' typologies and perception on SASs.

| SL | Adaptive Efficacy | Ecocentric (n = 355) | | Worried (n = 97) | | Anthropocentric (n = 48) | |
|---|---|---|---|---|---|---|---|
| | | Mean | STD | Mean | STD | Mean | STD |
| 1 | Afforestation | 4.3 | 0.7 | 4.2 | 0.8 | 2.8 | 0.9 |
| 2 | Focusing on weather forecast | 3.3 | 0.7 | 3.1 | 0.7 | 3.0 | 0.6 |
| 3 | Conserving soil | 4.2 | 0.6 | 4.2 | 0.6 | 3.3 | 0.6 |
| 4 | Migration of farmers | 3.0 | 0.9 | 3.1 | 0.9 | 2.3 | 0.6 |
| 5 | Changing planting time and practices | 3.2 | 0.8 | 3.2 | 0.8 | 2.8 | 0.5 |
| 6 | Using organic fertilizers | 4.1 | 0.7 | 4.1 | 0.6 | 3.7 | 0.9 |
| 7 | Replacing the crop varieties | 4.2 | 0.7 | 4.1 | 0.8 | 3.1 | 1.0 |
| 8 | Increasing irrigation | 4.1 | 0.7 | 4.1 | 0.7 | 3.1 | 0.4 |
| 9 | Mango Farming | 4.1 | 0.7 | 4.2 | 0.7 | 3.1 | 0.3 |
| 10 | Increasing employment | 2.9 | 0.9 | 3.0 | 0.9 | 3.6 | 0.8 |
| 11 | Livestock farming | 4.2 | 0.6 | 4.2 | 0.6 | 3.0 | 1.3 |
| 12 | Changing the seeding materials | 3.0 | 0.8 | 2.9 | 0.7 | 2.5 | 0.8 |
| 13 | Storing more water | 3.0 | 0.8 | 3.0 | 0.7 | 3.6 | 0.7 |
| 14 | Public awareness | 4.3 | 0.8 | 4.3 | 0.7 | 3.1 | 0.6 |
| 15 | Preparing for extreme weather | 3.6 | 0.7 | 3.5 | 0.6 | 3.7 | 0.6 |
| 16 | Control-diseased ecosystems | 3.6 | 0.6 | 3.7 | 0.7 | 3.0 | 0.7 |
| 17 | Change of disease and insect pests | 2.5 | 0.5 | 2.5 | 0.5 | 2.3 | 0.7 |
| 18 | Plant-adapted species | 3.6 | 0.6 | 3.7 | 0.7 | 3.5 | 0.8 |
| 19 | Adoption of high-yielding varieties | 3.6 | 0.6 | 3.6 | 0.6 | 3.2 | 0.9 |
| 20 | Changing of cropping pattern | 3.6 | 0.6 | 3.6 | 0.6 | 3.1 | 0.8 |

Pearson's correlation was conducted to show the bivariate relationship of the top SASs among all farmers' typologies (Table 5). The correlation was made with typologies' standardized K-means cluster value. The results reveal that 'Ecocentric' farmers have a significantly negative relationship with most of the SASs compared to other cluster farmers. By contrast, 'Worried' farmers have a significantly positive relationship with those SASs practices. 'Anthropocentric' farmers have a moderately negative relationship with most SASs practices and less interest in adopting those SASs.

**Table 5.** Pearson's correlation between best SASs and farmers typologies.

| | Ecocentric (*n* = 355) | Worried (*n* = 97) | Anthropocentric (*n* = 48) |
|---|---|---|---|
| Afforestation | −0.99 * | 0.76 | −0.48 |
| Conserving soil | −0.99 | 0.80 | −0.53 |
| Using organic fertilizers | −0.99 | 0.80 | −0.54 |
| Replacing the crop varieties | −1.00 * | 0.75 | −0.46 |
| Increasing irrigation | −0.99 | 0.80 | −0.53 |
| Mango Farming | −0.98 | 0.85 | −0.60 |
| Livestock farming | −0.99 | 0.80 | −0.53 |
| Storing more water | 0.99 | −0.81 | 0.54 |
| Public awareness | −0.99 | 0.80 | −0.53 |
| Control-diseased ecosystems | 0.81 | −0.99 | 0.88 |
| Plant-adapted species | −0.81 | 0.99 | −0.89 |
| Adoption of high-yielding varieties | −0.99 | 0.80 | −0.54 |

\* Correlation is significant at the 0.05 level (2-tailed).

## 4. Discussion

It is always challenging to make appropriate adaptation measures in agriculture to cope with climate change without an understanding of farmers' perceptions of climate change [21,53]. This is especially true for Bangladesh due to its substantial geopolitical role in food security. Subsequently, insight into the environmental attitudes of Bangladeshi farmers is crucial. As the warming climate, rainfall, groundwater resources, and extreme weather events pose a crucial threat to vital agricultural crop production in the study area, the adoption effective SASs to cope with these changes is urgently needed. There are huge challenges, as farmers believe in climate change and adaptive behaviors vary significantly from each other, which implies the necessity of demarcating farmers into various types based on their environmental perceptions and climate change. This study adopted a typology approach and revealed different types of farmers, such as Ecocentric, Worried, and Anthropocentric. Though there are some drawbacks to the typology approach, when considering those types [25,54], such outcomes may be inferred to aid rural farmers, agricultural extension officers, and decision-makers to inspire farmers to adopt SASs in their agricultural crop production practices. With this background, the effect of farmers' socioeconomic features (e.g., off-farming practices, size of land, and availability of modern technologies) should be examined in detail for further study.

The 'Anthropocentric' group of farmers perceived enough awareness to help them take appropriate environmental adaptations [18,31,55]. Interestingly, most farmers in the 'Ecocentric' and 'Worried' categories have recognized similar interests in SASs, which were also reported in some other studies [10,21,26,56–58]. We found that environmental values are a key factor for 'Ecocentric' and 'Anthropocentric' farmers but not 'Worried farmers'. A possible explanation for this result is related to the inherent sense of the economic values of these farmers, which prohibit their awareness of the environment. In fact, a high sense of environmental behavior can influence farmers' attitudes. Additionally, cultural diversity can affect one's environmental attitudes [24]. Though environmental knowledge had no substantial impact on pro-environmental behaviors, it is driven by environmental attitudes and environmental attitudes [59].

Divergent to 'Worried', the other two clusters have pro-environmental adaptive behavior that help them to adopt the SASs. Farmers in the 'Worried' category represented

the lowest environmental behavior due to their low educational status and age level, while 'Ecocentric' and 'Anthropocentric' farmers are motivated by pro-environmental values. This pro-environmental behavior is disapproved of for being the root of ecological disasters [60], where the value of nature initially aids human wellbeing. Therefore, financial motivations to adopt SASs may gain more economically interested attention from anthropocentric farmers, who are also considered as profit-induced adopters by several researchers, such as [18,31]. The concept of "sustainable intensification" was introduced instead of economic gain, which should be inspired by the anthropocentric farmer [30].

Government or non-government programs are concurrently supporting the assistances for the environment and farmers' economic drives. A low carbon agriculture plan (CAP) may benefit both farmers in the categories of 'Ecocentric' and 'Anthropocentric'. When considering 'Ecocentric' farmers, this CAP may have the main aim to decrease the environmental effects of activities including decreasing carbon emission, lower the cost of carbon, and reducing brickfields. Similarly, for the 'Anthropocentric' farmers, the accessibility of credit to invest in the farming practice could be the key to utilizing this government or non-government program. Additionally, the sustainable rural development program is a perfect example of Bangladeshi rural farmers which can attract much attention from both farmers' categories [60].

These aforementioned programs may not be beneficial for 'Worried' farmers, perhaps due to a low level of knowledge about these attempts. Farmers in the 'Worried' group have the highest non-agricultural income and the highest level of perceived risk, which may indicate a high probability of adopting SASs [30]. By contrast, farmers in the 'Worried' category showed the lowest level of environmental value. This outcome may be associated with the low level of educational qualification of this type of farmer. A study conducted by [61] showed the effect of educational level on environmental values. To improve this condition, agricultural incentives and rural development projects should have been implemented with this type of farmers to notify them about the benefit of the SAS adoption.

Interestingly, farmers in the 'Ecocentric' and 'Worried' clusters have the highest average in terms of the eight SASs adopted in this study. Farmers in this cluster had the highest percentage of SASs, such as afforestation and public awareness, and the second-highest SASs, including livestock and mango farming. The 'Anthropocentric' cluster had the highest average of SASs, e.g., using organic fertilizers. These results are in good agreement with the earlier works on environmental attitudes [31], where the environmental behavior of both clusters has been demonstrated to be pro-active. Based on these findings, it can be said that environmental values also affect farmers' adaptive behavior. Hence, related policies and programs concerning SASs should be prioritized to link the environmental benefits of SASs because farmers have various drives concerning environmental issues.

In fact, farmers' adaptive behaviors are highly influenced by their deeper knowledge of environmental issues, as [9,18,37,59,61] reported that interviewees with less interest in the environment than economic value were considered apathetic farmers. Furthermore, in comparison with the other two clusters, the "Worried" cluster showed a high sense of perceived risk. Farmers in the 'Ecocentric' group had a low sense of awareness of climate change, differentiating them from the other clusters. This low awareness of climate change also differentiated them from the ecologist cluster pioneered by Hyland [30]. Surprisingly, both ecocentric and anthropocentric farmers showed a significant negative association with SASs, while the worried category exhibited a significantly positive association with SASs [62]. Therefore, a comprehensive adaptation plan by taking the salient features of these three types of farmers into account is essential for enhancing SASs' efficacy.

There are some limitations to this study, as the current study relies on respondent responses. To conduct a more robust analysis, various indicators such as cultural beliefs, knowledge, traditions, biodiversity, and environmental awareness should be considered in further study. Second, future studies should explore the integrated mechanism of farmers' perceived adaptive efficacy and pro-environmental adaptation behavior in large-scale farming. Third, separate examinations of small farmers and larger farmers are

suggested, particularly at large farm levels. More importantly, this study is based on farmers' perceptions, and it may vary significantly from farmer to farmer, and it may even vary within the same respondents over time. However, such a kind of study is still advantageous in many aspects. On the whole, the outcomes of this research can be utilized to accelerate efficient policy implications to enhance the adoption of sustainable adaptive measures, as this study identifies the adaptive measures with their efficacies based on farmers' perceptions.

## 5. Conclusions

Bangladesh, as an agrarian country, must have a comprehensive understanding of farmers' typologies based on environmental issues and climate change in order to promote SASs and improve their resilience. Therefore, the present study focused on farmers' typologies based on environmental attitudes and their implication about SASs for northern Bangladesh. This research revealed the salient features of three types of farmers in northern Bangladesh. The findings suggest that understanding farmer's motivations for environmental values and climate change can help SAS supporters to choose the right information to convey the significance of adopting the SASs to farmers. Based on the farmer's profile, it was found that farmers in the 'Ecocentric' cluster had more experience, educational status, and the highest agricultural income, indicting pro-environmental adaptation behavior, and that higher educational levels can lead to strong environmental and climatic understanding.

Government and non-government organizations can take initiatives according to the farmers' typologies, aiming to improve their understanding of the most effective SASs. As a result, they may adopt SASs that will ultimately help them to cope with climate changes and extreme weather events and to contribute to ensuring food security.

**Supplementary Materials:** The following are available online at https://www.mdpi.com/article/10.3390/cli9120167/s1, Table S1: List of the 20 SASs and climate change statements

**Author Contributions:** Conceptualization, A.R.M.T.I., M.H. and M.J.; methodology, M.H.; software, M.J.; validation, G.M.M.A., E.A. and J.M.; formal analysis, M.H.; investigation, M.A.S.; resources, A.R.M.T.I.; data curation, M.J.; writing—original draft preparation, A.R.M.T.I. and J.M.; writing—review and editing, G.M.M.A.; visualization, A.R.M.T.I.; supervision, A.R.M.T.I. and K.T.; project administration, A.R.M.T.I.; funding acquisition, E.A.; J.M. and K.T. All authors have read and agreed to the published version of the manuscript.

**Funding:** This research was supported by Prince of Songkla University and the Ministry of Higher Education, Science, Research and Innovation, Thailand, under the Reinventing University Project (Grant Number REV64001). The authors also extend their appreciation to the Deanship of Scientific Research at King Khalid University for funding this work through Research Group under grant number (R.G.P.2 /194/42).

**Institutional Review Board Statement:** The study was conducted according to the guidelines of the Declaration of Helsinki, and approved by the Institutional Review Board (or Ethics Committee) of Dept. of Disaster Management, Begum Rokeya University, Rangpur (protocol code 2021/1000(2) and 15 April 2021.

**Informed Consent Statement:** Written informed consent has been obtained from the participants to publish this paper.

**Data Availability Statement:** Data are available on reasonable request on corresponding author.

**Acknowledgments:** We would like to thank the participants of the study regions, northern Bangladesh who cordially involved in this work.

**Conflicts of Interest:** The authors declare no conflict of interest.

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
