# Peer review of "Assessing Farmers’ Typologies of Perception for Adopting Sustainable Adaptation Strategies in Bangladesh"

_climate, doi:10.3390/cli9120167_

Round 1

Reviewer 1 Report

Title:

1/ condensate the main discovery into a short claim (full sentence, don't omit the verb), make sure the Title is short, original, groundbreaking, attractive, significant and easy to understand for everybody (never use abbreviations or technical terms)

2/ better reflect our international audience of readers, do not indicate limited impact

Abstract:

3/ better explain what motivated your work and why do you find the investigation of the research hypothesis so significant that it should be published on international level

4/ follow the established schema: A/ motivation + research hypothesis; B/ methods + results; C/ conclusions and interdisciplinary implications

5/ it is hard to find something new or unexpected, highlight the scientific novelty and quantify the economic importance of your discovery, clarify how will humanity benefit from your work

Introduction:

6/ avoid reference overkill by breaking down all clusters of references (use only 1 reference per claim/sentence)

7/ you can use Bangladesh as a case study, however, the manuscript should not miss the global point of view, review the topic from international point of view and refer to the latest problems related to environmental issues (refer to paper "Economic impacts of soil fertility degradation by traces of iron from drinking water treatment")

8/ build your research hypothesis more clearly (straightforward and groundbreaking claim that is confirmable or refutable) at the end of the Introduction chapter, justify the urgency of its investigation from global point of view; explain how will our readers benefit from your work

Materials and Methods:

9/ do not present encyclopedic data, present only reproducible methods

10/ check all typos ("oc" = °C etc.)

11/ our readers should find here (only and exclusively) detailed description of all your procedures (step by step, describe each apparatus and method used), anybody who reads this chapter should be able to repeat your methods and obtain exactly the same results

12/ do not refer to any local names, this brings nothing to our international audience of readers

13/ each Fig. and Tab. should be provided with detailed caption that will explain A/ what can be seen; B/ why is it important and C/ how is it related to the research hypothesis (explain the meaning of colors)

Discussion:

14/ show more self-criticism to your methods, discuss all limitations of your results

15/ more intensely compare your findings with existing literature, discuss industrial implication mentioned in paper "Green Entrepreneurship: Literature Review and Agenda for Future Research"

16/ take your research to the next level, provide a deeper synthesis of your results and reveal the mechanisms that shape them, this will allow you to uncover original theoretical insights

Conclusions:

17/ make sure you are presenting only new theoretical (globally applicable) findings that originate firstly from your work and are not deducible from other literature

Author Response

We have provided the response of the reviewer comments in the attached file

Reviewer 2 Report

The research question is interesting. However the paper is very poorly written. There are many issues regarding grammar that render the paper confusing and hard to read. The review report include more specific questions and suggestions. 

Author Response

(The authors gave the same response as above.)

Reviewer 3 Report

This paper reports on the results and analysis of a survey of farmers in Bangladesh with regard to their perceptions of climate change, environmental issues, and their implementation of sustainable adaptation strategies (SASs). The analysis uses PCA to identify a typology of three types of farmers based o their attitudes to climate change and the environment: ecocentric, worried and anthropocentric. The paper discusses these typologies and what they might mean for implementation of SASs

I think the research is valuable and novel. It is about farmers' views and has a focus on what this means for implementation of adaptation - an issue that is so important for places like Bangladesh but one in which top-down approach can still dominate. However, I think the presentation of the research and results could be improved to make the findings and their importance clearer. Therefore, I have recommended reconsider after major revisions. Key issues to be addressed are discussed below.

The paper needs editing to improve the language, and I think this would improve the clarity significantly. I appreciate the authors may not have English as a first language, so my criticisms are not focused on this. My comments below focus on where I think the intellectual content needs improving - but I do acknowledge that some of these criticisms might be due to my misunderstanding of the intended meaning.

Introduction

The introduction generally has all the necessary content and background for the paper. There's clearly a general introduction to the issue (1st paragraph), a focus on the need for and description of PAS (2nd paragraph) highlighting that some have productivity benefits. However, lines 68 to 74 are a little unclear: I took it to mean that economic incentives are the common mechanism to get farmers to change practices (especially in Europe) but that this doesn't seem to work in Bangladesh. I think the language just needs clarifying there.

The third paragraph talks about the perceptions of farmers, how they can be grouped into a typology and how this can be useful. It then describes previous studies that have done this, and why this one builds on that work. I think the importance of typologies can be strengthened in this section. You return to this later (I think) in the discussion, but it would be good to explain the benefits of typologies for policy and implementation. They can be really useful (despite some criticism) and potentially reveal important factors to focus on (exactly as you do here). Some further detail and explanation would strengthen the importance and usefulness of the paper.

Methods

Generally the method is strong but it is lacking some important details that would really improve the understanding of the paper.

The attitudinal statements need greater explanation. Looking at Table 1, it isn't clear where these came from. Were the farmers asked about each of these. For example, were farmers actually asked about "Air pollution"? They were clearly asked to rate something on a 5-point Likert scale but it isn't clear to me what they were asked to rate. Was it impact on their crops? Level of concern generally? Relation to climate change? etc. 

Relatedly, it isn't very clear to me what was asked about the SASs and the link between these are the attitudinal satements. The sentences around line 158 seem important here, but I can't quite follow them:

"We got 18 attitudinal statements after performing principle component analysis (PCA). It should be noted that the farmers have frequently adopted these sustainable adaptation strategies (SASs)."

This seems to suggest that the PCA was run on the rating of the SASs? Or at least a link between the SASs and the attitudinal statements. Later you look at the different SASs in each group which risks a circular logic. It needs to be clearer both what you asked the farmers about the SASs (as well as how the list was generated - you imply in the introdcution that it comes from an FAO list? I suggested just clarifying that in the methods). Were farmers asked to identify which they had done? Or were they asked to rate effectiveness of SASs? It is possible I have misunderstood the study and analysis! If so, I apologise, but I think greater clarity in the methods would help other readers.

I suggest that some more details on the survey and survey questions would help to clarify this, as well as provide opportunities for repeatability.

I'm not enough of an expert in statistics to fully assess the statistical analysis, but it looks rigorous and is clearly an important part of the paper.

Results

The paper would benefit from some more explanation of the three types identified.

The ecocentric type is described as having "high environmental value" - but not related to cliamte change. Some might criticise the separation of climate change and environmental values (climate change is an environmental issue). However, I think it is valid and understandable here if it is made clearer what is meant by "high environmental value"; I think these farmers care about the environment generally (even if they lack knowledge about climate change). Linking this back to the methods and the different statements might help - Table 1 implies that attitudes to deforestation and natural factors are important here.

The 'Worried' category is identified as (line 245) "representing the highest sense of perceived risk that can affect negatively in many regions of the world" and line 247 "less interest in environmental values as they have the most interest in perceived risk which suggests a lower likelihood to adopt sustainable practices". It isn't clear what the perceived risk is to (or from) - I think perceived risk to their crops/productivity from climate change? If so, I'm not sure why this would suggest a lower likelihood to adopt sustainable practices - I would expect greater perceived risk would encourage this?

Section 3.3: The discussion around the SAS needs to be clearer. In reporting the SASs, it isn't always clear if the farmers ares asked about whether they have implemented the SAS or whether they think it is effective (or both). Table 4 gives 'adaptive efficacy' but this is potentially what farmers think is effective but not necessarily what they themselves have tried. This has implications for the typology. Section 3.3 talks about farmer's 'interest' in SAS, which is might be about what they think will work or what they are willing to do. Later in section 3.3 (line 293) you say "Some SASs has a lower practice due to the features of the farmers.", implying that you have asked about what is being practiced. The differences in the attitudes of different types of farmers to the different SASs will then become clearer.

Discussion

The discussion provides a reasonably good summary of the results, although it is sometimes a little unclear (e.g. line 266 "‘Anthropocentric’ group farmers perceived enough awareness" Perceived awareness? Or had awareness? And awareness of what?). There is also some repetition - the importance of environmental values crops up a lot.

Some statements I don't think follow on from the analysis or are too general. For example: line 339 "Farmers in ’Worried’ represented the lowest environmental behavior due to their low educational status" - the results suggested that there was more to it than educational status and I that explanation seems to simplistic to me (many communities with low education show strong environmental values due to e.g. cultural beliefs). It might be the case here of course, but I'm not sure the results are that definitive. Either way, if it is important, then this suggests that education should be part of a comprehensive adaptation plan - something the authors could potentially include in the discussion?

Line 340: "This behavior is disapproved for being the root of ecological disasters [61] due to the anthropocentrism is a viewpoint where the value of nature initially gives human wellbeing." It isn't clear what behaviour is being talked about.

Conclusions

Line 396: "‘Ecocentric’ cluster farmers exhibited the higher experience, educational status, and the highest agricultural income indicting pro-environmental adaptive behavior, and the higher educational levels can trigger in higher environmental and climatic knowledge": this last part about climatic knowledge seems to contradict line 239: "‘Ecocentric’ indicates that this class of people perceived high environmental value with a low sense of awareness of climate change and perceived risk" Unless you mean climactic knowledge to relate to climate generally and not cliamte change?

Overall I think there is a really interesting study in this paper but the clarity of presentation needs improving. Some of this is language, and that can be improved with editing (maybe a professional editor if possible). However, some key clarifying around the methods (exactly what the surveys asked people and the link between attidudinal statements and the SASs) and clarification around what is meant by some key terms, especially environmental values and perceived risk. If these issues are addressed then I think the paper will be significantly clearer and the quality of the study and results will become much more evident.

Author Response

(The authors gave the same response as above.)

Round 2

Reviewer 1 Report

acceptable

Author Response

We have attached the reviewer comments response

Reviewer 3 Report

The authors have made some significant revisions to the paper in line with the reviewers comments, and overall the paper is definitely improved. However, the paper still needs some clarity around key issues. As such I suggest it needs another round of revisions.

Some of this is definitely still just language. The language has been improved, but in many places it still lacks some clarity. In some cases it is fairly obvious what is meant (e.g. line 47: "which are likely to decrease agricultural" - which are likely to decrease?) and these aren't too crucial. However, in other cases it affects the meaning considerably (e.g. line 72: "Government and non-government organizations have been supported by the farmers"; do you mean farmers have been supported by government and NGO?). These are just two examples. Given the short timeframes for reviewing demanded by the journal, I don't have the time to highlight every phrase of concern.

Title

I'm not sure on the phrase "perceived typologies". I think this could mean the typologies that farmers perceive. I think it should be something like "typologies of perception"

Introduction

This has been improved. I still think there is scope to highlight the benefits of a typology. The importance of typologies is mentioned a couple of times

Line 80: "The categorization of farmers’ typology-based perception is suitable to enhance the insights into the factors that affect the implementation of adaptation strategies [27]."

And:

Line 106: "Understanding farmer's perceived typologies on the environmental problem and climate change are necessary for implementing sustainable adaptive measures to enhance the farmer's ability to adapt at the farm level."

But this isn't really explained. I guess the question that isn't answered is 'how' does a typology "improve farmers' adaptive efficacy to climate change" (Line 103).

There are several other confusing sentences:

Line 78: "...based on their perception of different typologies towards climate change." It is their perception of climate change, not their perception of typologies of climate change, I think?

Line 88: "Nevertheless, the role of the environmental issue on farmer’s perceived typologies is still less investigated, due possibly to their new addition in the existing literature" It isn't clear what 'their' is in this sentence - environmental issues? Typologies?

Line 94: "For instance, an examination was held on the farmers’ perception and focusing on non-economic factors such as land ownership, the change creation, and recognition that affect farmers’ behavior which motivates the categorization of typologies [33]" This needs clarifying - what did the the examination find and why did it motivate typologies?

Line 96: "Over the several decades, this hypothesis has acted as the base of many published cited works..." It isn't clear what the hypothesis is here - I think it is that economic factors are key drivers of farmer behaviour change, but that discussion appears a lot earlier. Or the hypothesis that non-economic factors are important?

Methods

I'm still unclear on some of the details of the methods.

Would it be possible to list the 20 SAS chosen, perhaps in supplementary material?

Line 156: How were farmers asked to rate each SAS? (Likert scale? Scoring? etc.)

Line 157: These next few steps are not clear to me. You selected 73 Statements about cliamte change? Where did these come from - the literature again? Then a PCA reduced these to 18? What data was the PCA based on? Then, line 161, you say that the 18 statements were "amended from the existing literature" (so what about the PCA on the 73 statements?)

Also, by "both" attitudinal statements (line 159) do you mean both sets of statements (one related to SASs and one related to climate change)?

It is a shame that the survey questions can't be included, this might help. However, a very clear explanation of the steps would help improve this. You could even consider a diagram showing the methodological steps?

Results

Line 252: I'm not sure what is meant by "Thus, farmers do believe in cluster 2 that environmental issues are presented artificially." I think this is just language but it needs to be clear. What is meant by presented artificially?

Line 343: I noted before that I thought the sentence "Farmers in ’Worried’ represented the lowest environmental behavior due to their low educational status while ‘Ecocentric’ and ‘Anthropocentric’ farmers are motivated by pro-environmental values" was too simplisitic and didn't reflect the results above. In fact, according to Table 3, the mean value for educational status was lower (and very similar) for Anthropocentric farmers who are motivated by environmental values, suggesting that education wasn't a key factor. I don't mind if the authors want disagree with this point, but it does need some explanation in light of their own results.

Line 345: "This pro-environmental behavior is disapproved for being the root of ecological disasters [60] t where the value of nature 346 initially gives human wellbeing." You added 'pro-environment' in response to me previous comment. I'm not sure that I agree that 'pro' environment behaviour is the root of ecological disasters. Not caring about the environment is the cause of ecological disasters surely? Again, maybe just a language issue, but this is a good example of where the meaning is wrong. It makes me unsure as to whether other statements are incorrect due to language issues.

Overall, I think the paper needs further revision - focusing on clarity of the methods and mainly on presentation. I'm very conscious that the authors might be writing in a second or third (or more) language, and I don't like criticising papers for language, but there are a number of cases in this where it is hard to understand what is meant or that comes across as incorrect. For this reason I've suggested minor changes, but I recommend that the authors conduct a detailed edit on the manuscript for language beyond only the most important issues highlighted above. Given more time, I would try to highlight other language concerns but don't want to hold up publication.

In my view, this is definitely an interesting and valuable paper.

Author Response

(The authors gave the same response as above.)
